# The spread of macroscopic droplets from a simulated cough with and without the use of masks or barriers

**Atreyus A. Bhavsar**[ID]*

The Blake High School, Minneapolis, Minnesota, United States of America

* aabhavsar22@blakeschool.org

## Abstract

One of the main challenges during the COVID-19 pandemic is the lack of safety measures and guidelines to reduce the risk of viral spread among people during gatherings. This study was conducted to evaluate the distance of oral and nasal droplet spread in a model that simulates coughing and sneezing in a public setting, specifically a school setting, to guide faculty and staff members with safety measures and guidelines to reduce droplet spread. Several models were prepared to observe and visualize the spread of fluid simulating respiratory droplets in places such as the classroom and the cafeteria, in which a student would be more susceptible to contract a virus since individuals cannot wear masks while eating. For all trials, a 2.54 centimeter balloon with 0.3 milliliters of diluted fluorescent paint was placed inside a mannequin head and was exploded outwards from the mannequin's mouth at 5 pounds per square inch (psi). Using a black light, the expelled fluorescent macroscopic droplets were visualized. When applying safety precautions and guidelines such as mandating face masks, the results of the experiments conducted in this study with a surgical mask, were extremely positive. However, without other safety precautions such as face masks and barriers, social distancing proved to be ineffective. In conclusion the most effective way to prevent droplet spread during activities where masks simply cannot be worn, such as eating, is to apply barriers between the individuals. Applying barriers and wearing masks successfully prevented macroscopic droplet spread.

## Introduction

Mitigation efforts aimed at tackling COVID-19 spread, with recommendations including covering a person's face with a mask, social distancing, and regular hand washing, appear to be effective [1]. This study was conducted to evaluate the distance of simulated respiratory droplet spread and the effectiveness of applying masks and barriers as a mitigation strategy to improve safety. Activities such as speaking, coughing, sneezing and even breathing produce oral and nasal droplets containing viral particles [2–5].

There appear to be two components of a sneeze or cough, a ballistic droplet component and a turbulent gas or puff component which have been visualized using high speed videography,

**Data Availability Statement:** All relevant data are within the paper.

**Funding:** The author received no specific funding for this work.

**Competing interests:** The authors have declared that no competing interests exist.

distortion of projected schlieren light beams and shadowgraph imaging [6–11]. The velocity of the cough airflows has been measured as high as 14 m/s [10, 12]. The horizontal distance of a gas cloud after a cough or sneeze may travel more than 8 meters and aerosol transport has been documented at a distance of 4 meters [13]. The size of particles ejected during a cough or sneeze ranges from 0.1 to 1000 microns [14]. Mathematical models have been used to calculate the effect of drag, diffusion, gravity, humidity, temperature and wind flow on the velocity and distances traveled by respiratory droplets [15].

While it was previously thought that larger droplets fall to the ground after a short distance, smaller yet macroscopic droplets may travel farther and be predominantly responsible for transmitting disease [16].

One model of respiratory droplet spread has involved the use of fluorescent dye in a small latex balloon which is inflated until it bursts [17–19]. The droplets produced have been visualized with an ultraviolet light [17–20]. This model has been used in multiple experiments to examine the spread of droplets in clinical and surgical settings, with and without masks and various barriers [17–21]. Surgical masks have been shown to alter and reduce the respiratory jets and droplet spread from coughs and sneezes [9, 19–23]. The experiments in this manuscript utilized the model of respiratory droplet spread described above.

Understanding the maximum spread of oral droplets may assist in producing more effective measures to combat COVID-19 and protect students and faculty when returning to school.

The CDC recommends that people should practice social distancing at least 6 feet apart in combination with other preventative measures such as wearing face masks [24]. However, the majority of people believe that only social distancing is effective in preventing the spread of COVID-19, leading them to believe that they are safe, when in fact they are not.

## Materials and methods

For each experiment three trials were conducted in a ventilated room simulating a classroom or cafeteria. The tables in experiments 3 through 6 were adjusted to mimic the exact table dimensions in the Blake High School cafeteria. The school table dimensions were 306.07 centimeters in length, 93.98 centimeters in width, and 76.2 centimeters in height. In addition, the chairs in the Blake High School cafeteria were 48.26 centimeters in height, and the chairs used in this experiment were the same height. For each trial only one mannequin head (M1) simulated a student coughing. To ensure non-contamination and efficiency between each experiment, tablecloths (Paper Art Co., Inc., Indianapolis, Indiana, U.S.A.,) clear plastic wraps (Polyvinyl Films, Inc., Sutton, MA, U.S.A,) and plastic bags (Fleet Farm, Brooklyn Park, MN, U.S.A.) were placed on the tables, mannequin heads (Florocraft, China,) and body and replaced after each experiment. In addition, three different colors of fluorescent paint (Testors Craft, Hawthorn Pkwy, Vernon Hills, IL, U.S.A.,) orange, green, and pink, were used for each trial with one color being used per trial. This would allow clear distinction between each trial run. To mimic the viscosity of saliva, 1mL of fluorescent paint was diluted with the same volume of water. A wooden frame was built around a mannequin mask (Creatology, China) to simulate a laryngeal cough from the interior, while also resembling a person's head. Plastic was attached between the wooden frame and the mask, as well as on the eyes, in order to ensure droplets were only exiting from the mouth and nasal cavity. The mannequin's mouth was cut in a circular shape and had a diameter of 5.08 centimeters, and the nostrils were cut the same way and had a diameter of 1.27 centimeters in order to simulate the measurements of a student's mouth and nostrils when coughing. These measurements were obtained from the author's mouth and nostrils as shown in Fig 1B, and the author has the body habitus of a typical high school student. Two screws were placed on the sides of the wooden frame in order to

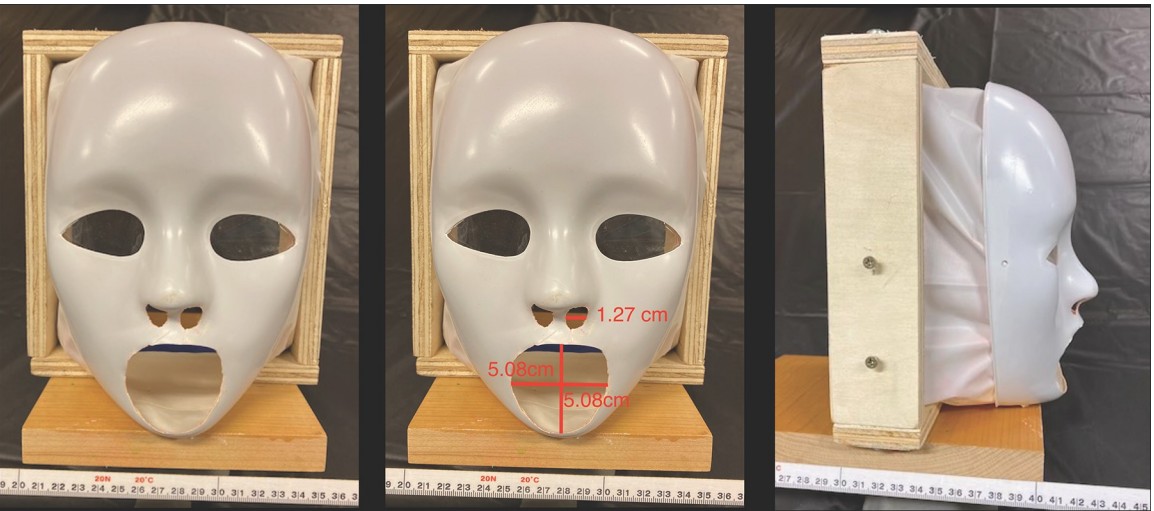

**Fig 1. Dimensions and different views of M1.** A white metric measuring tape with centimeter and millimeter markings is shown in each figure. (A) The front of the mannequin head is displayed. The head consists of a wooden base and frame as well as plastic connected to the mask and frame. (B) The measurements of the nostrils and mouth are shown. (C) The side view of M1 is displayed as well as two screws which were used to simulate the human ear to anchor the mask.

attach the ear loops of a mask to mimic the top and bottom points of an ear as shown in Fig 1C. Depending on the simulation, whether the mannequin was sitting down or standing up, respectively, two or three identical boxes (Fellowes, U.S.A) covered with plastic bags which were replaced between each experiment, were placed underneath the wooden frame of the mannequin mask as well as other mannequin heads. These boxes were not only used to support the head of the mannequins but also to simulate the body of a student. All the heights of the mannequins were based off of the measurements of the author when sitting at a height of 134.62 centimeters and standing at a height of 177.8 centimeters. To simulate the cough, an air compressor (Bostitch, U.S.A.) was used to fill a 2.54 centimeter long latex balloon (Toysmith, Taichung, Taiwan) filled with 0.3mL of the diluted fluorescent paint, a reasonable volume of fluid that one might expel when coughing, at the mouth of the mannequin mask and inflated at 5 psi until it burst [17]. Five psi has been previously reported as the pressure of a laryngeal cough [18, 25]. After each trial, a black light (Vansky, China) was used in order to observe and record the spread of droplets as shown in Fig 2A–2C. A white metric measuring tape with centimeter and millimeter markings (Lufkin, U.S.A.) was used to measure the distance of droplet spread.

The purpose of the first six experiments was to simulate the spread of droplets from a student's unprotected cough while seated in a cafeteria setting. The last three experiments were designed to study the spread of droplets and the effectiveness of a mask when standing in an open space such as a hallway.

The first experiment was designed to determine the maximum spread of droplets traveling straight outward from M1. As shown in Fig 3B, M1 was seated, measuring 134.62 centimeters tall, at one far end of the table. The length of the table was 306.07 centimeters.

Similarly, the second experiment was designed to measure the lateral distance of droplet spread. As shown in Fig 4B, M1 was placed in the middle of the long side of the table which was 459.70 cm in length. M1 was placed at the middle of the table to measure the maximum spread of droplets traveling to both sides. The maximum distance of droplet spread on both sides of M1 were recorded.

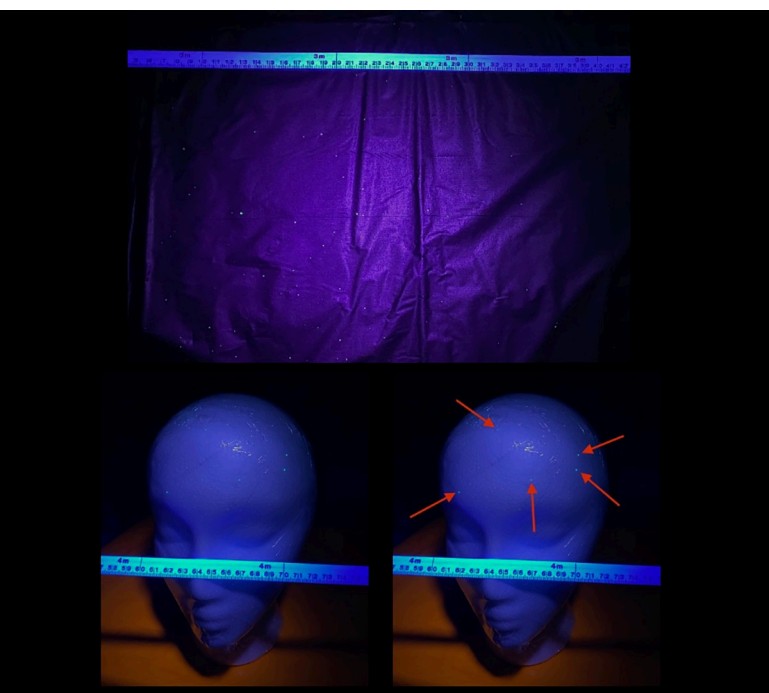

**Fig 2. Droplets on surrounding surfaces.** A white metric measuring tape with centimeter and millimeter markings is present in each figure. (A) Fluorescent paint droplets can be visualized on the table by illumination with a black light. (B) Five fluorescent paint droplets were found on the forehead of one of the mannequins surrounding M1. (C) Arrows point to five fluorescent paint droplets on the head of the mannequin.

For the third experiment, 10 mannequins were oriented in normal eating positions without social distancing measures to examine the spread of droplets. A reasonable distance to simulate normal seating positions of students in a cafeteria was by orienting them 25.40 centimeters shoulder width apart. All mannequins were aligned symmetrically. Going clockwise from M1, the mannequins were named in consecutive numbers, as shown in Fig 5B. The distance from M1's mouth to the other mannequins' bodies was 53.34 centimeters to M2 as well to M10, 114.94 centimeters to M3 and M9, 168.91 centimeters to M4 and M8, 121.92 centimeters to M5 as well as to M7, and 93.98 centimeters to M6.

The purpose of the fourth experiment was to identify the maximum height of droplet spread on the barrier as well as on the top cover of the barrier to determine a sufficient barrier height and whether a top cover was necessary in order to prevent droplets from spreading to the surrounding mannequins and table. To determine whether or not droplets could travel

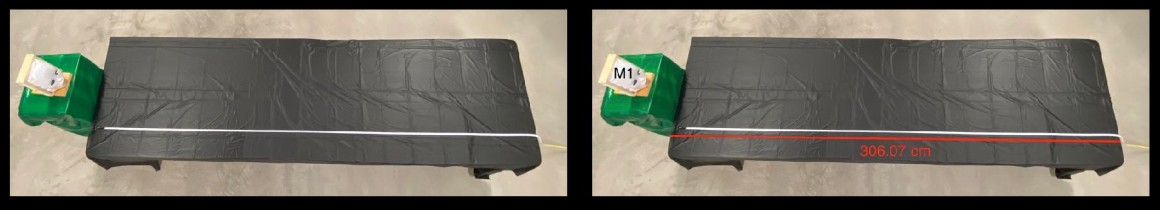

**Fig 3. Frontal droplet spread when sitting simulation.** A white metric measuring tape with centimeter and millimeter markings is displayed in each figure. (A) A bird's eye view photo of the full table in front of the mannequin simulating the cough is shown. (B) The exact measurement of the table is labeled, and the mannequin is labeled as M1.

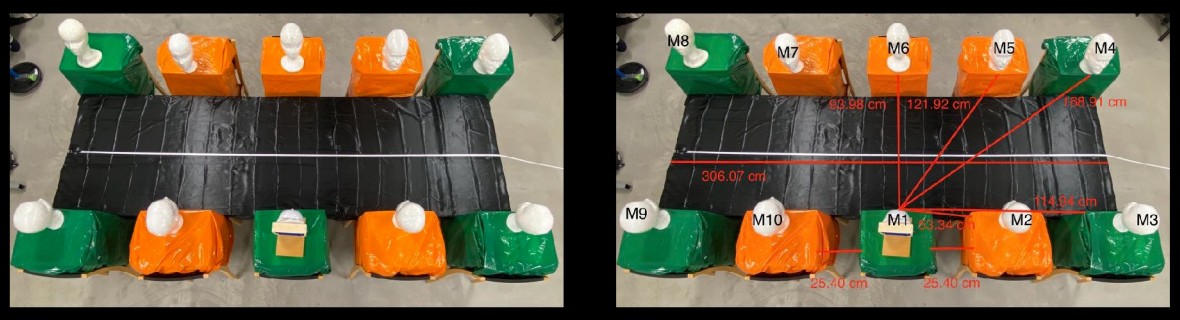

**Fig 4. Lateral droplet spread when sitting.** A white metric measuring tape with centimeter and millimeter markings is shown in each figure. (A) A bird's eye view photo of the mannequin simulating the cough, placed in the middle of a long table, is displayed. (B) The measurement of the length of the table is shown and the mannequin at the center is labeled M1.

over the barrier, plexiglass was placed over the top of the barrier. Similar to the third experiment, the nine other surrounding mannequins were seated in normal eating positions with a barrier around M1, as shown in Fig 6B. Three 91.44 centimeter tall white boards were used as the barrier surrounding M1. One of the white boards was placed in front of M1, and the two adjacent whiteboards were placed on the sides, ending at the edge of the table. Plexiglass was applied over the top of the barrier to determine if droplets could travel over the barrier. As shown in Fig 6D, a comfortable eating space for a student was determined to be 60.96 centimeters by 38.10 centimeters while still keeping the barriers as close as possible to reduce the dispersion of droplets.

In the fifth experiment, the positions of each mannequin remained constant. The two side walls of the barrier were extended off the edge of the table by 17.78 centimeters while the eating space remained the same, as shown in Fig 7. Similar to the previous experiment, the purpose of this experiment was to examine whether droplets would spread anywhere outside of the barrier.

The purpose of the sixth, seventh, and eighth experiments was to simulate the spread of droplets from a person coughing without a mask, with a mask worn improperly, as shown in Fig 8A and 8B, and with a mask worn properly, as shown in Fig 8C and 8D while standing in an open space such as a hallway. These experiments were designed to test the effectiveness of a surgical mask by observing whether or not macroscopic droplets were found anywhere beyond the mask. First, table cloths were placed on the floor, covering a large area surrounding the mannequin. Then, three identical boxes were placed on a chair to mimic the height of a student. The mannequin head was then placed on top of the boxes, reaching a height of 5ft 10".

**Fig 5. Measurements of normal seating positions.** A white metric measuring tape with centimeter and millimeter markings can be seen in each figure. (A) A bird's eye view photo of 10 mannequins seated in normal positions at a table can be seen. These seating positions were designed to mimic students sitting at a cafeteria table without safety protocols. (B) All mannequins are labeled as well as the measurements of the distances from the mouth of M1 to the bodies of M2, M3, M4, M5, and M6. The shoulder-to-shoulder distance between M1 to M2 as well as M1 to M10 is displayed. The length of the table is also shown.

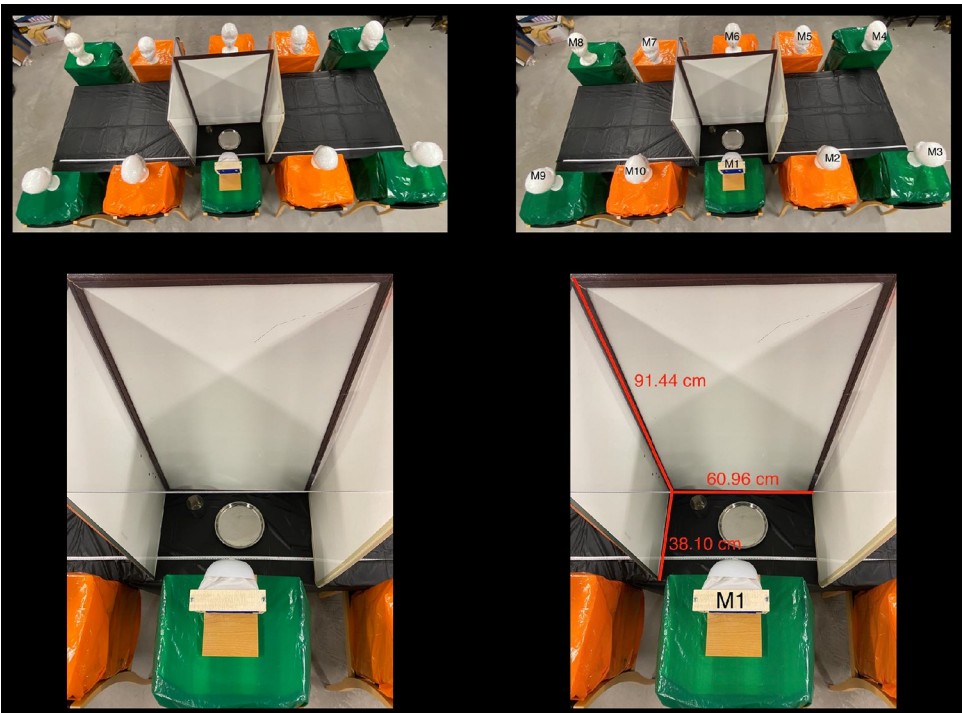

**Fig 6. Measurements of normal seating positions and barrier.** A white metric measuring tape with centimeter and millimeter markings is shown in each figure. (A) A bird's eye view photo of 10 mannequins sitting in the exact same positions as experiment 3 is shown. Three whiteboards were positioned in front of the middle mannequin on the bottom of the photo. A plate and a cup were placed on the interior of the whiteboards in front of that mannequin to better simulate the eating space. In addition, plexiglass can be seen on top of the whiteboards. (B) The ten mannequins are labeled. (C) A closer picture of the barrier in front of the mannequin simulating the cough is shown as well as a plate and a cup. There is also a metric ruler in from of the mannequin on the table as a scale. (D) The height, length, and width of the barrier is shown, and the mannequin simulating the cough is labeled M1.

After each trial a black light was used to examine the droplets on the ground and the data was recorded. In the seventh experiment, as shown in Fig 8A and 8B, the surgical mask was placed below the nose and only covered the mannequin's mouth. In eighth experiment, the mask was fitted around the nose and mouth of the mannequin, as shown in Fig 8C and 8D.

Droplet sizes were measured in experiments 3, 4, and 5 with a digital fractional caliper (Ironton, China) with an accuracy of 0.013 millimeters. A total of 67 droplets were measured. The diameter of the largest droplets ranged from 0.95 mm to 3.57 mm, while the diameter of the smallest droplets ranged from 0.22 mm to 0.91 mm. The majority of the droplets were less than 1 mm in diameter.

## Results

The results are separated by experiment type, and the distances of droplet spread are listed in centimeters.

The purpose of the first experiment was to examine the maximum distance of frontal droplets spread. The maximum distance of droplets traveling straightforward from M1 was 272.42 centimeters with a range of 181.61 centimeters to 272.42 centimeters as shown in Table 1.

In experiment two, which was designed to determine the maximum lateral dispersion of droplets, the maximum distance of droplets traveling to the left of M1 was 200.66 centimeters with a range of 170.18 centimeters to 200.66 centimeters. Similarly, the maximum distance of

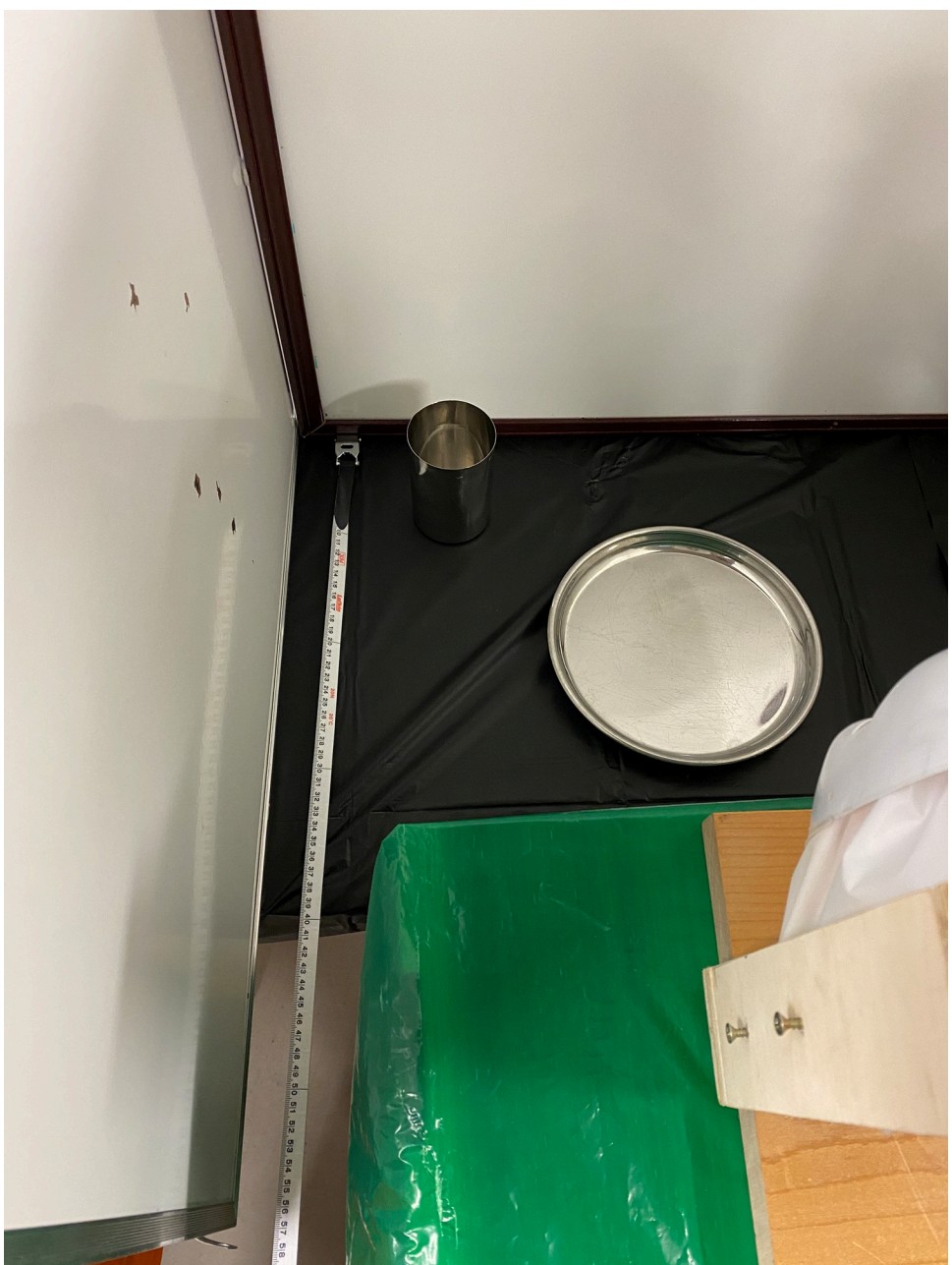

**Fig 7. Measurements of extended barrier.** The side walls of the barrier have been extended off the edge of the table. There is a white metric measuring tape with centimeter and millimeter markings that travels from the mannequin's eating space off the edge of the table. A plate and cup are also present.

droplets traveling to the right of M1 was 184.79 centimeters with a range of 158.75 centimeters to 184.79 centimeters as shown in Table 2.

In experiment three, which was designed to examine the spread of droplets in a cafeteria or classroom setting at a table, macroscopic droplets were found on every mannequin as well as in all of their eating spaces, which was within at least 38.10 centimeters from the mannequin. Droplets were found on the bodies and heads of the surrounding mannequins as shown in Table 3.

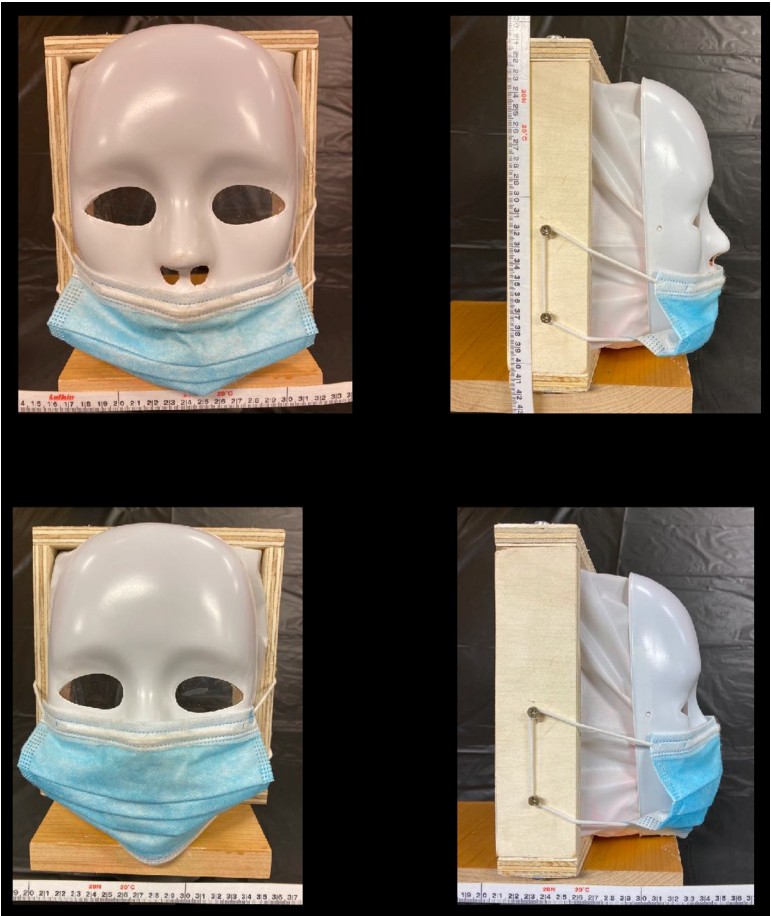

**Fig 8. Frontal and side view of M1 wearing a mask in different positions.** A white metric measuring tape with centimeter and millimeter markings is positioned in each figure. (A) The Frontal view of M1 wearing a surgical mask improperly is displayed. (B) The side view of M1 wearing a surgical mask improperly is shown. (C) The frontal view of M1 wearing a surgical mask properly is presented. (D) The side view of M1 wearing a surgical mask properly can be seen.

The fourth experiment was designed to determine the height of droplet spread on the barrier, as well as whether droplets spread on the inside of the top cover and anywhere outside the barrier. As shown in Table 4, the maximum height of droplet spread on the barrier from the fourth experiment was 90.50 centimeters, with a range of 88.70 centimeters to 90.50 centimeters. Droplets were also found on the inside of the top cover of the barrier in each trial. Droplets were most dense on the board at the height of the mouth of M1 and were found farther

**Table 1. (Experiment 1): Maximum distance of droplet spread traveling straightforward on the table.**

|  | Maximum distance of droplet spread (centimeters) |
|---|---|
| Trial #1 | 181.61 |
| Trial #2 | 204.47 |
| Trial #3 | 272.42 |

This table shows the measurements of the maximum distance of frontal droplet spread during a simulated cough that landed on the table.

**Table 2. (Experiment 2): Maximum distance of droplet spread traveling laterally on the table.**

| | Maximum distance of droplet spread to the left of M1 (centimeters) | Maximum distance of droplet spread to the right of M1(centimeters) |
|---|---|---|
| Trial #1 | 200.66 | 184.79 |
| Trial #2 | 176.53 | 132.72 |
| Trial #3 | 170.18 | 158.75 |

This table displays the measurements of the maximum droplet spread travelling laterally from M1.

spread apart the higher they were found on the board. Droplets were found on the body of M2 and M10 in the third trial.

The fifth experiment was designed to test whether droplets were found on surfaces other than the extended barrier. As shown in Table 5, there were no droplets found on the body or head of any mannequin as well as in their eating spaces with the use of the extended barrier.

The sixth, seventh, and eighth experiment were designed to observe the effectiveness of a surgical mask in preventing droplet spread. As shown in Table 6, droplets were found at a maximum radius of 248.92 centimeters with a range of 232.41 centimeters to 248.92 centimeters when coughing standing up without a surgical mask. In the eighth experiment, droplets were found at a maximum radius of 119.38 centimeters with a range of 99.06 centimeters to 119.38 centimeters when wearing a mask improperly. In the final experiment, no droplets found beyond the mask when the mask was worn properly. In experiment 6, 7, and 8, the results of droplet spread without a mask, with a mask worn improperly (covering only the mouth, but not the nose), and with a mask worn properly (covering both the mouth and nose) were deduced as follows. The finding that there were droplets found on the inner surface of the surgical mask (the inside surface of the mask facing the nose and mouth) during each separate trial in experiment 6 both with the mask worn improperly and properly but not in the space or surfaces around M1 with the mask worn properly, led to the deduction that the mask prevented droplet spread.

## Discussion

As schools begin plans to reopen, safety precautions and guidelines need to be established to protect students and teachers from respiratory droplet spread in order to mitigate transmission and infection from COVID-19. The experiments conducted in this study help clarify some characteristics of macroscopic droplet spread which can aid in the implementation of safety measures.

**Table 3. (Experiment 3): Droplet spread in normal cafeteria seating positions.**

| | Mannequins on which droplets were found | Droplets found on the head of the mannequin | Droplets found on the body of the mannequin | Droplets found in the eating space (within 38.10 centimeters from mannequin) of the surrounding mannequins |
|---|---|---|---|---|
| Trial #1 | M5, M6, M7, M8, M10 | M5, M6, M7 | M5, M6, M7, M8, M10 | M2, M3, M4, M5, M6, M7, M10 |
| Trial #2 | M2, M5, M6, M7, M8, M10 | M2, M5, M6, M7, M8, M10 | M6, M7, M8, M10 | M2, M5, M6, M7, M8, M9, M10 |
| Trial #3 | M2, M4, M5, M6, M7, M10 | M2, M4, M5, M6, M10 | M2, M4, M5, M6, M7, M10 | M2, M3, M4, M5, M6, M7, M9, M10 |

This table presents the mannequins on which droplets were found as well as the eating spaces on which droplets were found.

**Table 4. (Experiment 4): Droplet spread with barrier use.**

| | Maximum height of droplet spread (centimeters) | Droplets found on the cover of the Barrier | Mannequins on which Droplets were Found with the Barrier | Droplets found on the head of the mannequin | Droplets found on the body of the mannequin | Droplets Found in the Eating space (within 38.10 centimeters from mannequin) of the Surrounding Mannequins with the Barrier |
|---|---|---|---|---|---|---|
| Trial #1 | 89.10 | Yes | None | None | None | None |
| Trial #2 | 90.50 | Yes | None | None | None | None |
| Trial #3 | 88.70 | Yes | M2, M10 | M2, M10 | M2, M10 | None |

This table shows the maximum height of droplets that traveled on the barrier, the inside of the top cover of the barrier, the surrounding mannequins, and their eating spaces.

It is not completely known whether respiratory macroscopic droplets can spread beyond 6 feet (2 meters) [22]. Although, a recently published study showed a simulated jet containing microscopic droplets traveling up to 12 feet [23], and another recent study show that turbulent gas clouds can travel 23–27 feet (7–8 meters) [13]. However, it is not proven at this time that microscopic respiratory jets are the predominant mechanism of disease transmission. In fact, macroscopic droplets maybe more likely the predominant mechanism involved in disease transmission since many outbreaks such the 1981 outbreak of infectious meningitis in a Texas elementary school involved students who became infected while seated within less than 3 feet of the first person infected. In this case one could hypothesize that it was the shorter traveling larger droplets that landed on the children which caused the infection, rather than microscopic droplets which are known to travel up to 12 feet (>2 meters) away or gas clouds travelling up to 23–27 feet (7–8 meters) away, where none of the other students were infected [26]. In the experiments contained in this manuscript, it became clear that social distancing at 6 feet (1.83 meters) alone was not effective at preventing macroscopic droplet spread. In multiple trials, sitting or standing without a mask or barrier, the maximum distance of macroscopic droplets was found farther than 6 feet (1.83 meters) from the mannequin simulating the cough. In the experiments simulating a cafeteria setting, droplets were found in the majority of the other mannequins' eating space. Although it may not seem harmful, if infectious droplets land in one's food, one could be infected. However, when applying a barrier, droplet spread was much more sparse. Furthermore, when extending the sides of the barrier 17.78 centimeters offset from the table's edge toward M1, the results showed a more protective effect. There was no droplet spread to any of the surrounding mannequins as well as anywhere outside the barriers. This would seem to provide protection from sideways spread of droplets that could travel to and infect a person sitting next to the person coughing.

**Table 5. (Experiment 5): Droplet spread with extended barrier use.**

| | Mannequins on which droplets were found | Droplets found in the eating space (within 38.10 centimeters from mannequin) of the surrounding mannequins |
|---|---|---|
| Trial #1 | None | None |
| Trial #2 | None | None |
| Trial #3 | None | None |

This table displays the effectiveness of the extended barrier in the fifth experiment.

**Table 6. (Experiment 6, Experiment 7, Experiment 8): Droplet spread with and without a surgical mask.**

| | Maximum Radius of droplet spread without wearing a mask (centimeters) | Maximum radius of droplet spread when wearing a mask improperly (centimeters) | Maximum radius of droplet spread when wearing a mask properly (centimeters) | Droplets found on the inside surface of the mask when wearing a mask improperly and properly | Droplets found on the outside surface of the mask when wearing a mask improperly and properly |
|---|---|---|---|---|---|
| Trial #1 | 232.41 | 119.38 | 0 | Yes | No |
| Trial #2 | 248.92 | 99.06 | 0 | Yes | No |
| Trial #3 | 236.86 | 109.86 | 0 | Yes | No |

This table shows the spread of droplets without wearing a mask, while wearing a mask improperly and wearing a mask correctly.

One of the weaknesses of the experiments that were conducted was that only three trials were conducted for each experiment, whereas a larger number of trials would have enabled a more sophisticated statistical analysis. In addition, the mannequins in each experiment were only facing straight, whereas human beings are constantly moving. Moreover, because all experiments were conducted in a ventilated room, droplet spread could have been influenced [27]. In all experiments the room ventilation emanated from central ducts in the ceiling on one side of the room with the vents facing M1 and all experiments were conducted with M1 facing the ventilation ducts. Thus, the airflow was directed towards M1 and could have decreased the distance of droplet travel. However, since the airflow was not specifically measured, it could have influenced the results in other unknown ways. Furthermore, an air compressor was used to inflate a balloon filled with fluorescent paint to simulate the cough. If a real human being was used during these experiments, the results may have been different. The viscosity of the fluorescent paint was diluted to mimic the viscosity of saliva. However, the viscosity of the solution or of saliva was not measured. There has been experimental evidence from other studies that the droplets generated in respiratory droplet models can vary in size and may be susceptible to evaporation [28]. Although the assessment of droplet spread on the surrounding surfaces around M1 was performed immediately after the simulated cough, it is possible that rapid evaporation could have led to an underestimate of any droplets that might have penetrated the mask and evaporated prior to observation of the droplets remaining on the surfaces around M1. However, that would presume that there would be no residual fluorescent residue remaining on the surrounding surfaces after potential evaporation. If the droplets did evaporate without any fluorescent residue, this may have led to an underestimation of droplet spread particularly if any droplets travelled beyond the mask in experiment 6. This could have produced a false negative finding. However, since the inside surface of the mask contained droplets and no droplets or fluorescent residue were visible on the outside of the mask or surrounding surfaces, it is less likely that substantial macroscopic droplets could have escaped the mask. In addition, it was not known whether a cloth mask or an N95 mask would perform differently in these experiments. However, despite these challenges, the strengths of these experiments include the generation of valuable information on droplet spread from a simulated cough. These experiments have consistently shown that macroscopic droplets travel farther than was previously thought. To the author's knowledge, this is the first study to report data on droplet spread using barriers in a cafeteria type setting. In addition, the information provided in this study can help guide mitigation efforts in preventing droplet spread. The prevention of droplet spread can help reduce transmission of the Coronavirus during this global pandemic [29].

## Conclusions

Based on the results of these experiments, social distancing at a distance of 6 feet (1.83 meters) without a mask or barrier was ineffective at preventing droplet spread. However, a surgical mask was effective at preventing droplet spread anywhere beyond the mask of the coughing mannequin.

Physical barriers should also be established in places where masks are not worn, such as cafeterias in schools to limit droplet spread. Clear barriers would seem prudent so that students would be less likely to lean back to see or to talk to other students. A barrier that was aligned with the edge of the table was not sufficient to prevent lateral droplet spread. Therefore, the barrier was extended to 17.78 centimeters beyond the edge of the table toward M1, and this prevented lateral droplet spread. In addition, droplets were consistently found on the top cover of the barrier, meaning that droplets could travel over the barrier. When the barrier extended past the edge of the table, in other words in front of the table 17.78 centimeters, it effectively prevented droplets from spreading anywhere outside the barrier. Based on these findings, it would be prudent for barriers to be constructed with this in mind.

Wearing a mask properly over the mouth and nose or utilizing barriers where masks cannot be worn, such as cafeterias, were effective in preventing macroscopic droplet spread.

Most importantly, this series of experiments can help guide schools to establish safety guidelines and precautions as they re-open to prevent droplet spread both in the classroom and in the cafeteria setting.

## Acknowledgments

I would like to thank Niharika A. Bhavsar, Nirayudh A. Bhavsar, and Abdhish R. Bhavsar for their invaluable assistance, expertise and guidance.

## Author Contributions

**Writing – review & editing:** Atreyus A. Bhavsar.

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
