## [Decision Letter · Decision Letter 0]

11 Nov 2020

PONE-D-20-26616

“The Spread of Macroscopic Droplets from a Simulated Cough with and without the Use of Masks or Barriers”

PLOS ONE

Dear Mr Bhavsar,

Thank you for submitting your manuscript to PLOS ONE. After careful consideration, we feel that it has merit but does not fully meet PLOS ONE’s publication criteria as it currently stands. Therefore, we invite you to submit a revised version of the manuscript that addresses the points raised during the review process.

Your manuscript was assessed by three external experts, whose comments are appended to this letter. 

Two of the reviewers raise the important point that your submission does not engage fully with the body of existing literature on droplet dissemination and how it is affected by mask-wearing. This is important in order that the reader can understand how your findings relate to what is already known in the literature. 

Reviewer 3 notes a key limitation with the methods you used that has not been discussed in your manuscript, namely that it is unclear whether the distribution of droplet sizes in your model 'cough system' is representative of those generated by humans.

Please respond to these criticisms in your response to reviewers, as well as the others provided by the reviewers, and update your manuscript accordingly.

We look forward to receiving your revised manuscript.

Kind regards,

Dr Joseph Donlan

Senior Editor

PLOS ONE

Journal Requirements:

Reviewers' comments:

Reviewer's Responses to Questions

**Comments to the Author**

1. Is the manuscript technically sound, and do the data support the conclusions?

Reviewer #1: Partly

Reviewer #2: No

Reviewer #3: Partly

2. Has the statistical analysis been performed appropriately and rigorously? 

Reviewer #1: N/A

Reviewer #2: No

Reviewer #3: N/A

3. Have the authors made all data underlying the findings in their manuscript fully available?

Reviewer #1: Yes

Reviewer #2: No

Reviewer #3: Yes

4. Is the manuscript presented in an intelligible fashion and written in standard English?

Reviewer #1: Yes

Reviewer #2: Yes

Reviewer #3: Yes

5. Review Comments to the Author

Reviewer #1: Very nice effort from what appears to be a high-school student - done in the family garage with the help of family members! I commend the effort - well done! See if the journal can waive their publication fees ($1350 at present) for a high-school student.

Some suggestions to improve this paper below:

- change all distances to metric - science is metric now

- combine the smaller figures into a 4-panel figure, and the larger ones into a 3-panel figure. Make sure the Figure legends fully describe what readers are seeing - without having to refer to the main text - each Figure and its legend should stand alone. Use arrows in the figures and refer to them in the figure legend to make things clearer - sometimes it is not quite clear what the authors is saying, e.g. the overhang of the vertical barriers over the edge of the table.

- expand your reference list - before claiming a first for anything, and I wouldn't do this anyway, as by the time your paper is published, other studies may have been published already (some may already be in review), making this statement inaccurate and redundant. It does not help your study get published, and in fact some reviewers may be more critical of your paper because of you claiming this.

Some additional related studies on coughing and airflow/droplet dissemination with/without masks:

https://journals.plos.org/plosone/article?id=10.1371/journal.pone.0034818

https://journals.plos.org/plospathogens/article?id=10.1371/journal.ppat.1003205

https://www.pnas.org/content/115/5/1081

https://www.ncbi.nlm.nih.gov/pmc/articles/PMC2843945/

https://www.nejm.org/doi/full/10.1056/NEJMicm0904279

https://pubmed.ncbi.nlm.nih.gov/18843121/

https://pubmed.ncbi.nlm.nih.gov/32982136/

https://pubmed.ncbi.nlm.nih.gov/32304605/

https://pubmed.ncbi.nlm.nih.gov/32631450/

The limitations are good.

Try to avoid fish-eye/panoramic photos for illustration as they may distort the distances - diagrams are also useful, but it would be helpful if the author can retake all of these photos with a ruler (metric) of some sort in the image as a scale.

The cafeteria table model looks too narrow? Can the authors provide some measurements of their typical school cafeteria table to validate this model more?

Reviewer #2: This paper is very useful in demonstrating the effects of wearing masks. However, it is not clear how the results were deduced. The authors should add a section showing some intermediate results. For example, adding photographs showing aerosol, particulate or even tracers motion for scenarios with and without mask will be more convincing.

Reviewer #3: This is an admirable study performed by a high school student, and the methods and results could serve as excellent educational material in school settings. However, in my opinion, the methods are not sophisticated enough for publication in a journal with the stature of Plos One. You should cite more of the literature on the transport of droplets, as much has been published lately, e.g., from Lydia Bourouiba from MIT in JAMA 2020 and William Lindsley in 2016. A major limitation of the methods is the inability to measure the particles sizes generated by your apparatus to know whether or not they are similar to those generated by humans. One point that may seem picayune is that scientific papers should include measurements using the metric system.

6. PLOS authors have the option to publish the peer review history of their article (what does this mean?). If published, this will include your full peer review and any attached files.

Reviewer #1: No

Reviewer #2: No

Reviewer #3: No

---

## [Author Response · Author response to Decision Letter 0]

29 Dec 2020

Response to Reviewers 

Reviewers’ Comments to the Author and the Author’s Responses:

Reviewer 1

1. Comment from Reviewer 1: Very nice effort from what appears to be a high-school student - done in the family garage with the help of family members! I commend the effort - well done! 

Author response: Thank you!

2. Comment from Reviewer 1: See if the journal can waive their publication fees ($1350 at present) for a high-school student.

Author response: Thank you! That would be very much appreciated.

3. Comment from Reviewer 1: “Change all distances to metric.”

Author response: All measurements have been changed to the metric system. 

4. Comment from Reviewer 1: “Combine the smaller figures into a 4-panel figure, and the larger ones into a 3-panel figure.” 

Author response: Thank you for pointing this out. I have combined the smaller figures into 4 panel figures and the larger ones into 2 panel figures.

5. Comment from Reviewer 1: “Make sure the Figure legends fully describe what readers are seeing - without having to refer to the main text - each Figure and its legend should stand alone.” 

Author response: The figure legends have been changed and fully describe the figures in the current manuscript. 

6. Comment from Reviewer 1: “Use arrows in the figures and refer to them in the figure legend to make things clearer”

Author response: Arrows have been used in some of the figures to make things clearer and are referred to in the figure legend.

7. Comment from Reviewer 1: “Expand your reference list.” 

Author response: The reference list has been expanded with more recent publications. The additional references that I reviewed have been included in references 3 through 15, 18 through 21, and 25. 

8. Comment from Reviewer 1: “before claiming a first for anything, and I wouldn't do this anyway, as by the time your paper is published, other studies may have been published already (some may already be in review), making this statement inaccurate and redundant. It does not help your study get published, and in fact some reviewers may be more critical of your paper because of you claiming this.”

Author response: I have reviewed the literature again six months after the original manuscript was written and I am still unable to find any publications regarding cafeteria barrier use as I have described in this manuscript.

9. Comment from Reviewer 1: “The limitations are good.”

Author response: Thank you!

10. Comment from Reviewer 1: “Try to avoid fish-eye/panoramic photos for illustration as they may distort the distances – diagrams are also useful, but it would be helpful if the author can retake all of these photos with a ruler (metric) of some sort in the image as a scale.”

Author response: All panoramic photos have been eliminated, and all photos have been retaken with a metric ruler. 

11. Comment from Reviewer 1: “The cafeteria table model looks too narrow? Can the authors provide some measurements of their typical school cafeteria table to validate this model more?”

Author response: The measurements of the school cafeteria have been validated and reproduced in each of the experiments as explained in lines 120 through 124.

Reviewer 2

1. Comment from Reviewer 2: “This paper is very useful in demonstrating the effects of wearing masks.”

Author response: Thank you!

2. Comment from Reviewer 2: “It is not clear how the results were deduced.”

Author response: The results were deduced by observing the fluorescent paint droplets that were ejected onto the surfaces surrounding the mannequin after the simulated cough event.

3. Comment from Reviewer 2: “The authors should add a section showing some intermediate results. For example, adding photographs showing aerosol, particulate or even tracer motion for scenarios with and without masks will be more convincing.”

Author response: I tried to capture the particle spread in motion but the particles were too small to capture by my video equipment. 

Reviewer 3

1. Comment from Reviewer 3: “This is an admirable study performed by a high school student, and the methods and results could serve as excellent educational material in school settings.”

Author response: Thank you! 

2. Comment from Reviewer 3: “However, in my opinion, the methods are not sophisticated enough for publication in a journal with the stature of Plos One.”

Author response: The methods of the simulated cough used in these experiments were previously used in manuscripts that have already been published in the New England Journal of Medicine, American Academy of Ophthalmology website, and the Canadian Journal of Ophthalmology in references 17, 18, and 19 in the current manuscript. Therefore, I respectfully submit that the methods that were used in this study deserve consideration for publication in Plos One. 

3. Comment from Reviewer 3: “You should cite more of the literature on the transport of droplets, as much has been published lately, e.g., from Lydia Bourouiba from MIT in JAMA 2020 and William Lindsley in 2016.”

Author response: Additional scientific publications including the ones suggested in the above comment have been reviewed and cited in the current manuscript in lines 72 through 110. 

4. Comment from Reviewer 3: “A major limitation of the methods is the inability to measure the particles sizes generated by your apparatus to know whether or not they are similar to those generated by humans.”

Author response: The experimental model used in the current manuscript has already been used in several other scientific published papers listed below. These have been cited as references 17, 18, and 19 in the current manuscript. In addition, the droplet sizes were measured during several experiments, and this is addressed in lines 518 to 522. Most of the droplets were less than 1 mm and that is consistent with droplet sizes that have been measured in the human cough as cited in reference 14 in the current manuscript. 

1. Canelli R, Connor CW, Gonzalez M, Nozari A, Ortega R. Barrier Enclosure during Endotracheal Intubation. N Engl J Med. 2020;382(20):1957-1958. doi: 10.1056/NEJMc2007589. 

2. Felfeli T, Batawi HI, Aldrees SS, Mandelcorn ED (2020, June 3). Infection Control Measures During Simulated Slit-Lamp Examination. Retrieved July 15, 2020, from https://www.aao.org/clinical-video/infection-control-measures-during-simulated-slit-l

3. Felfeli T, Batawi HI, Aldrees SS, Hatch W, Mandelcorn ED Utility of patient face masks to limit droplet spread from simulated coughs at the slit lamp. Can J Ophthalmol; 2020;55(5):e163-e165. doi: 10.1016/j.jcjo.2020.06.010. 

5. Comment from Reviewer 3: “One point that may seem picayune is that scientific papers should include measurements using the metric system.”

Author response: All measurements in the current manuscript have been converted to the metric system.

---

## [Decision Letter · Decision Letter 1]

27 Jan 2021

PONE-D-20-26616R1

The Spread of Macroscopic Droplets from a Simulated Cough with and without the Use of Masks or Barriers

PLOS ONE

Dear Dr. Bhavsar,

Thank you for submitting your manuscript to PLOS ONE. After careful consideration, we feel that it has merit but does not fully meet PLOS ONE’s publication criteria as it currently stands. Therefore, we invite you to submit a revised version of the manuscript that addresses the points raised during the review process.

We look forward to receiving your revised manuscript.

Kind regards,

Zezhi Li, Ph.D., M.D.

Academic Editor

PLOS ONE

Reviewers' comments:

Reviewer's Responses to Questions

**Comments to the Author**

1. If the authors have adequately addressed your comments raised in a previous round of review and you feel that this manuscript is now acceptable for publication, you may indicate that here to bypass the “Comments to the Author” section, enter your conflict of interest statement in the “Confidential to Editor” section, and submit your "Accept" recommendation.

Reviewer #1: (No Response)

Reviewer #2: (No Response)

2. Is the manuscript technically sound, and do the data support the conclusions?

Reviewer #1: Partly

Reviewer #2: No

3. Has the statistical analysis been performed appropriately and rigorously? 

Reviewer #1: N/A

Reviewer #2: No

4. Have the authors made all data underlying the findings in their manuscript fully available?

Reviewer #1: Yes

Reviewer #2: No

5. Is the manuscript presented in an intelligible fashion and written in standard English?

Reviewer #1: Yes

Reviewer #2: Yes

6. Review Comments to the Author

Reviewer #1: Almost there.

Figure 2 first image - the tape measure figures are not visible properly - you cannot include any blurred images/dimensions in a scientific paper as it raise doubts about the results/interpretation - as well as the ability for others to reproduce the result accurately. Please retake this image - ensuring that the tape figures are clear and visible.

I note the concerns of one other reviewer about the method - please cite these two papers in the appropriate place to counter other similar criticisms like this:

https://royalsocietypublishing.org/doi/10.1098/rsif.2009.0388.focus

https://royalsocietypublishing.org/doi/10.1098/rsif.2009.0311.focus

Reviewer #2: Appears not yet revised thoroughly following:

This paper is very useful in demonstrating the effects of wearing marks. However, it is not clear how the results were deduced. The authors should add a section showing some intermediate results. For example, adding photographs showing aerosol, particulate or even tracers motion for scenarios with and without mask will be more convincing.

7. PLOS authors have the option to publish the peer review history of their article (what does this mean?). If published, this will include your full peer review and any attached files.

Reviewer #1: **Yes: **Julian W Tang

Reviewer #2: No

---

## [Author Response · Author response to Decision Letter 1]

14 Feb 2021

Response to Reviewers 

Dear Dr. Li,

Thank you for providing me the opportunity to submit a revised version of the manuscript. “The Spread of Macroscopic Droplets from a Simulated Cough with and without the Use of Masks or Barriers” for publication in the journal of the Public Library of Science. I appreciate the time and effort that you and the reviewers dedicated to address the limitations in the paper and provide feedback. All of the changes suggested by the reviewers have been updated and addressed in the revised manuscript. 

Reviewers’ Comments to the Authors:

Reviewer 1

Comment from Reviewer 1: “Almost there.”

Author response: Thank you!

Comment from Reviewer 1: “Figure 2 first image - the tape measure figures are not visible properly - you cannot include any blurred images/dimensions in a scientific paper as it raise doubts about the results/interpretation - as well as the ability for others to reproduce the result accurately. Please retake this image - ensuring that the tape figures are clear and visible.”

Author response: The first image in Figure 2 has been retaken so that the measuring tape is clear and visible. 

Comment from Reviewer 1: “I note the concerns of one other reviewer about the method - please cite these two papers in the appropriate place to counter other similar criticisms like this.”

Author response: The method of how the results were deduced regarding the experiments with masks and the associated shortcomings have been addressed in lines 294 to 300, 339 to 344, and 347 to 359. References 27 and 28 have been revised with the reviewer’s suggested new references added to the manuscript.

Reviewer 2

Comment from Reviewer 2: “This paper is very useful in demonstrating the effects of wearing masks.”

Author response: Thank you!

Comment from Reviewer 2: “It is not clear how the results were deduced.”

Author response: The method of how the results were deduced regarding the experiments with masks and the associated shortcomings have been addressed in lines 294 to 300.

Comment from Reviewer 2: “The authors should add a section showing some intermediate results. For example, adding photographs showing aerosol, particulate or even tracer motion for scenarios with and without masks will be more convincing.”

Author response: I tried to capture the particle spread in motion but the particles were too small to capture by my video equipment. I did add an explanation of how the methods were deduced in lines 294 to 300, and I added columns to table 6 explaining that droplets were present on the inside of the mask and not on the outside of the mask during the simulations. 

Sincerely,

Atreyus A. Bhavsar

---

## [Decision Letter · Decision Letter 2]

3 Mar 2021

PONE-D-20-26616R2

The Spread of Macroscopic Droplets from a Simulated Cough with and without the Use of Masks or Barriers

PLOS ONE

Dear Dr. Bhavsar,

Thank you for submitting your manuscript to PLOS ONE. After careful consideration, we feel that it has merit but does not fully meet PLOS ONE’s publication criteria as it currently stands. Therefore, we invite you to submit a revised version of the manuscript that addresses the points raised during the review process.

We look forward to receiving your revised manuscript.

Kind regards,

Zezhi Li, Ph.D., M.D.

Academic Editor

PLOS ONE

Reviewers' comments:

Reviewer's Responses to Questions

**Comments to the Author**

1. If the authors have adequately addressed your comments raised in a previous round of review and you feel that this manuscript is now acceptable for publication, you may indicate that here to bypass the “Comments to the Author” section, enter your conflict of interest statement in the “Confidential to Editor” section, and submit your "Accept" recommendation.

Reviewer #1: All comments have been addressed

Reviewer #2: (No Response)

2. Is the manuscript technically sound, and do the data support the conclusions?

Reviewer #1: Yes

Reviewer #2: No

3. Has the statistical analysis been performed appropriately and rigorously? 

Reviewer #1: Yes

Reviewer #2: No

4. Have the authors made all data underlying the findings in their manuscript fully available?

Reviewer #1: Yes

Reviewer #2: No

5. Is the manuscript presented in an intelligible fashion and written in standard English?

Reviewer #1: Yes

Reviewer #2: Yes

6. Review Comments to the Author

Reviewer #1: The author has responded satisfactorily to my comments - congratulations! For interest - some links to the use of barriers and masks in schools in Southeast Asia - where they have dealt with the COVID-19 pandemic much more effectively:

Thailand:

https://www.google.co.uk/amp/s/metro.co.uk/2020/08/10/school-children-put-boxes-return-class-lockdown-13108120/amp/

https://www.google.co.uk/amp/s/www.dailymail.co.uk/news/article-8611385/amp/Thai-kindergartners-sealed-perspex-boxes-playtime-fight-against-coronavirus.html

Hong Kong:

https://www.google.co.uk/amp/s/amp.scmp.com/news/hong-kong/education/article/3084267/masks-while-singing-no-basketball-and-recess-inside-hong

Taiwan:

https://www.google.co.uk/amp/s/www.cbc.ca/amp/1.5505031

Vietnam:

https://www.google.co.uk/amp/s/www.vox.com/platform/amp/21270817/coronavirus-schools-reopen-germany-vietnam-new-zealand

For interest - here are some links showing actual barriers used in sine schools in Southeast Asia - where they have managed the COVID-19 pandemic much more effectively:

Reviewer #2: Appears that the following have not yet been revised thoroughly:

This paper is very useful in demonstrating the effects of wearing masks. However, it is not clear how the results were deduced. The authors should add a section to show some intermediate results. For example, adding photographs showing the aerosol, particulate or even tracers motion for scenarios with and without mask will be more convincing.

1. Not clear what have been revised, photographs are still not too clear.

2. There are articles published recently using high-speed camera with higher quality photographs.

7. PLOS authors have the option to publish the peer review history of their article (what does this mean?). If published, this will include your full peer review and any attached files.

Reviewer #1: **Yes: **Julian W Tang

Reviewer #2: No

---

## [Author Response · Author response to Decision Letter 2]

15 Mar 2021

Response to Reviewers 

I received the response dated March 3 regarding my manuscript submission. In my prior responses to the reviewers and my revisions submitted on February 14, I have responded to each of the reviewers' comments and made all the requested and appropriate revisions to the manuscript. 

Sincerely,

Atreyus A. Bhavsar

Reviewers’ Comments to the Authors:

Reviewer 1

1. Comment from Reviewer 1: “The author has responded satisfactorily to my comments - congratulations!”

Author response: Thank you!

2. Comment from Reviewer 1: “For interest - some links to the use of barriers and masks in schools in Southeast Asia - where they have dealt with the COVID-19 pandemic much more effectively”

Author response: Thank you!

Reviewer 2

1. Comment from Reviewer 2: “This paper is very useful in demonstrating the effects of wearing masks.”

Author response: Thank you!

2. Comment from Reviewer 2: “It is not clear how the results were deduced.”

Author response: The method of how the results were deduced regarding the experiments with masks and the associated shortcomings have previously been addressed in lines 294 to 300 during the prior round of revisions.

3. Comment from Reviewer 2: “The authors should add a section showing some intermediate results. For example, adding photographs showing aerosol, particulate or even tracer motion for scenarios with and without masks will be more convincing.”

Author response: I tried to capture the particle spread in motion but the particles were too small to capture by my video equipment. I previously added an explanation of how the methods were deduced in lines 294 to 300, and I added columns to table 6 explaining that droplets were present on the inside of the mask and not on the outside of the mask during the simulations. 

4. Comment from Reviewer 2: “Not clear what have been revised, photographs are still not too clear.”

Author response: The revisions are clearly listed in my previous response to reviewers in lines 294-300. In the last set of revisions, Figure 2.A was the only figure that was requested to be revised and this was revised to show the measuring tape in the background. 

5. Comment from Reviewer 2: “There are articles published recently using high-speed camera with higher quality photographs.”

Author response: I do not have access to a high-speed camera, and the images that have been submitted have already been approved by both Reviewer 1 and Reviewer 2 in prior revisions of this manuscript.

---

## [Decision Letter · Decision Letter 3]

24 Mar 2021

PONE-D-20-26616R3

The Spread of Macroscopic Droplets from a Simulated Cough with and without the Use of Masks or Barriers

PLOS ONE

Dear Dr. Bhavsar,

Thank you for submitting your manuscript to PLOS ONE. After careful consideration, we feel that it has merit but does not fully meet PLOS ONE’s publication criteria as it currently stands. Therefore, we invite you to submit a revised version of the manuscript that addresses the points raised during the review process.

One of the reviewer's comments should be carefully addressed.

We look forward to receiving your revised manuscript.

Kind regards,

Zezhi Li, Ph.D., M.D.

Academic Editor

PLOS ONE

Reviewers' comments:

Reviewer's Responses to Questions

**Comments to the Author**

1. If the authors have adequately addressed your comments raised in a previous round of review and you feel that this manuscript is now acceptable for publication, you may indicate that here to bypass the “Comments to the Author” section, enter your conflict of interest statement in the “Confidential to Editor” section, and submit your "Accept" recommendation.

Reviewer #2: (No Response)

2. Is the manuscript technically sound, and do the data support the conclusions?

Reviewer #2: No

3. Has the statistical analysis been performed appropriately and rigorously? 

Reviewer #2: No

4. Have the authors made all data underlying the findings in their manuscript fully available?

Reviewer #2: No

5. Is the manuscript presented in an intelligible fashion and written in standard English?

Reviewer #2: Yes

6. Review Comments to the Author

Reviewer #2: Without clear photographs as said, what is the value of this paper ?

However, in order not to cause too much trouble, perhaps the authors can improve the paper by trying the following, if they do not have access to a high-speed camera :

1. Draw better diagrams on what they try to present, and put those diagrams next to the unclear photographs.

2. Compare with other references showing clear photographs.

7. PLOS authors have the option to publish the peer review history of their article (what does this mean?). If published, this will include your full peer review and any attached files.

Reviewer #2: No

---

## [Author Response · Author response to Decision Letter 3]

31 Mar 2021

Response to Reviewers 

Dear Dr. Li,

Thank you for providing me the opportunity to submit a revised version of the manuscript. “The Spread of Macroscopic Droplets from a Simulated Cough with and without the Use of Masks or Barriers” for publication in the journal of the Public Library of Science. I appreciate the time and effort that you and the reviewer dedicated to address the limitations in the paper and provide feedback. All of the changes suggested by the reviewers have been addressed below.

Reviewer’s Comments to the Author:

Reviewer 2

1. Comment from Reviewer 2: “Without clear photographs as said, what is the value of this paper ? However, in order not to cause too much trouble, perhaps the authors can improve the paper by trying the following, if they do not have access to a high-speed camera : 1. Draw better diagrams on what they try to present, and put those diagrams next to the unclear photographs. 2. Compare with other references showing clear photographs.”

Author response: I sent all the native tiff figure files to the academic editor and he responded as follows: 

“Dear Dr. Bhavsar, 

The pictures you send to me are very clear. Thank great! You can submit these pictures as supplementary files if you possible. 

Kind regards, 

Zezhi Li 

Academic Editor 

PLOS ONE”

The editorial office also wrote the following to explain that the built PDF images are of low resolution and the original figure files that I uploaded were of high resolution: 

“Dear Dr. Li, 

Thank you for acting as Academic Editor for this submission to PLOS ONE. Further to your previous correspondence with the Corresponding Author, we have received the email below along with figures attached. 

Unfortunately, we do sometimes find that the figure quality in the PDF for review can be poor, but please be assured that the submission PDF is for reviewing purposes only and does not reflect the appearance of the final publication. Should the manuscript be accepted, the figures will be typeset at the resolution of the original submitted figures. 

I have attached a copy of the Revision 3 PDF for review to this email and you can access to the original source files via the blue download link in the top right corner of the PDF rendering of the figure. 

If I can be of any assistance in forwarding a response to the Corresponding Author on your behalf, please do let me know and I would be happy to help. If you respond directly to the corresponding author, please could I ask that you copy us in at - plosone@plos.org. 

Please do not hesitate to contact me if you have any queries, or if I can be of any assistance at all. Many thanks again for your time and efforts with this manuscript. 

Kind regards, 

Noelle Noelle Gibbs 

Editorial Office”

Therefore, the academic editor, Zezhi Li, has asked me to submit the manuscript again without needing any additional revisions to the figures since the original figures submitted are of high resolution. 

Sincerely,

Atreyus A. Bhavsar

---

## [Editor Report · Decision Letter 4]

5 Apr 2021

The Spread of Macroscopic Droplets from a Simulated Cough with and without the Use of Masks or Barriers

PONE-D-20-26616R4

Dear Dr. Bhavsar,

We’re pleased to inform you that your manuscript has been judged scientifically suitable for publication and will be formally accepted for publication once it meets all outstanding technical requirements.

Kind regards,

Zezhi Li, Ph.D., M.D.

Academic Editor

PLOS ONE
---

## [Editor Report · Acceptance letter]

12 Apr 2021

PONE-D-20-26616R4 

The Spread of Macroscopic Droplets from a Simulated Cough with and without the Use of Masks or Barriers 

Dear Dr. Bhavsar:

I'm pleased to inform you that your manuscript has been deemed suitable for publication in PLOS ONE. Congratulations! Your manuscript is now with our production department. 

Kind regards, 

on behalf of

Dr. Zezhi Li 

Academic Editor

PLOS ONE